# A Comprehensive Review on the Isolation, Bioactivities, and Structure–Activity Relationship of Hawthorn Pectin and Its Derived Oligosaccharides

**DOI:** 10.3390/foods13172750

**Published:** 2024-08-29

**Authors:** Tao Li, Wenhua Ji, Hongjing Dong, Yingqun Wu, Lanping Guo, Lei Chen, Xiao Wang

**Affiliations:** 1Key Laboratory for Applied Technology of Sophisticated Analytical Instruments of Shandong Province, Shandong Analysis and Test Center, Qilu University of Technology (Shandong Academy of Sciences), Jinan 250014, China; litaosd@qlu.edu.cn (T.L.);; 2Key Laboratory for Natural Active Pharmaceutical Constituents Research in Universities of Shandong Province, School of Pharmaceutical Sciences, Qilu University of Technology (Shandong Academy of Sciences), Jinan 250014, China; 3Guizhou Ecological Food Creation Engineering Technology Center, Guizhou Medical University, Guizhou 550025, China; 4College of Food Science and Technology, Guangdong Ocean University, Zhanjiang 524088, China

**Keywords:** hawthorn pectin-derived oligosaccharides, extraction techniques, functional characteristics, structure–activity relationship, functional foods

## Abstract

Hawthorn (*Crataegus pinnatifida* Bunge) has been highlighted as an excellent source of a variety of bioactive polymers, which has attracted increasing research interest. Pectin, as a kind of soluble dietary fiber in hawthorn, is mainly extracted by hot water extraction and ultrasonic or enzymatic hydrolysis and is then extensively used in food, pharmaceutical, and nutraceutical industries. Numerous studies have shown that hawthorn pectin and its derived oligosaccharides exhibit a wide range of biological activities, such as antioxidant activity, hypolipidemic and cholesterol-reducing effects, antimicrobial activity, and intestinal function modulatory activity. As discovered, the bioactivities of hawthorn pectin and its derived oligosaccharides were mainly contributed by structural features and chemical compositions and were highly associated with the extraction methods. Additionally, hawthorn pectin is a potential resource for the development of emulsifiers and gelling agents, food packaging films, novel foods, and traditional medicines. This review provides a comprehensive summary of current research for readers on the extraction techniques, functional characteristics, structure–activity relationship, and applications in order to provide ideas and references for the investigation and utilization of hawthorn pectin and its derived oligosaccharides. Further research and development efforts are imperative to fully explore and harness the potential of hawthorn pectin-derived oligosaccharides in the food and medicine fields.

## 1. Introduction

Hawthorn (*Crataegus pinnatifida* Bunge) is mainly distributed in East Asia, North America, and Europe [1]. It is important to note that hawthorn, as a “longevity food”, is rich in a variety of nutrients and has been approved by the Ministry of Health of China as the raw material of functional food. In recent years, accumulating evidence on the function of hawthorn has mainly focused on the pharmacological effects of flavonoids, polyphenols, and organic acids [2,3,4,5]. It is reported to be a multifunctional fruit, which has a variety of effects, such as antioxidants, weight loss, aiding digestion, and the prevention of diabetes mellitus, hypertension, atherosclerosis, and so on [6]. Indeed, the multiple functions of hawthorn benefit from its abundance of functional ingredients and bioactive compounds. Hawthorn fruit contains about 6.4% pectin, and the rich pectin substance determines hawthorn’s unique processing performance and products [7]. The growing interest in medical and functional food from hawthorn pectin, a natural product, is a result of its multifunctional activities and better accessibility compared with other cultivated fruits.

Pectin is a naturally occurring macromolecular polysaccharide found mainly in the middle lamella of higher plants, which accounts for up to 30% of the dry mass of the primary cell wall [8]. Chemically, the backbone of pectin consists of D-galacturonic acid linked by α-1, 4-glycosidic bonds with varying degrees of methyl esterification. Depending on the mode of covalent or ionic interactions, they are mainly classified into four types of structural domains: homogalacturonic acid (HG), rhamnogalacturonic acid type I (RG I), rhamnogalacturonic acid type II (RG II), and xylogalacturonic acid (XGA) (Figure 1A) [9,10]. Degradation of pectin by cutting glycosidic bonds using specific pectinases results in small molecules of oligosaccharides (Figure 1B) [11,12], which have attracted scientists’ intense research interests. Current studies have shown that pectin and its derived oligosaccharides rely on its molecular weight, esterification degree, viscosity, and other structural features to confer different biological activities, including antioxidant, anti-inflammatory, anti-cancer, hypoglycemic, and hypolipidemic effects in vitro and in vivo [13,14,15,16]. The compositional structure and functional activity of pectin vary depending on the variety, maturity, storage period, and extraction process [17]. Usually, pectin rises in the fifth week after the fruit sets until the stone is formed, and then the pectin content decreases, leading to fruit softening because of enzymatic degradation and solubilization of protopectin [18]. The availability of pectin increases with the increase in storage time [19]. Natural pectin comes from a wide range of sources, and different contents of pectin exist in fruit and vegetable raw materials such as pomegranate, lemon, apple, grapefruit, hawthorn, cocoa husk, sunflower heads, sugar beet, pumpkin, watermelon, pears, and potato pulp [20,21]. Hawthorn pectin exhibits high molecular weight, high viscosity, stable rheological properties, excellent biocompatibility, and is quite stable under acidic conditions in the stomach. Compared with commercially available pectin from citrus, lemon, and apple, hawthorn pectin has higher hardness, gumminess, and chewiness due to its high content of galacturonic acid (GalA) and degree of esterification (DE), making it easy to form gels [22,23]. Hawthorn pectin-derived oligosaccharides display various physicochemical properties, such as being low calorie, having low viscosity, having high stability, being safe, and being non-toxic. Due to their complex chemical structure and diversity of functional groups, hawthorn pectin and its derived oligosaccharides exhibit an extensive array of biological activities, which are commonly believed to be promising biopolymers with a wide range of applications.

Recently, the demand for functional and multipurpose food has increased, and the focus on health has shifted from treatment to prevention [24]. The available properties of hawthorn pectin and its derived oligosaccharides can fulfill the requirements of functional foods and dietary supplements while also providing preventive benefits against certain diseases in everyday life [1,17]. Despite the numerous studies on hawthorn pectin and its derived oligosaccharides, there is a lack of a comprehensive review of the subject, especially on hawthorn pectin-derived oligosaccharides. Therefore, this paper aims to systematically review the recent literature on hawthorn pectin and its derived oligosaccharides, covering the extraction methods and functional characteristics. Additionally, the underlying mechanism of bioactivities and the structure–activity relationship were also discussed and explored, with further emphasis on the current and prospective applications of hawthorn pectin and its derived oligosaccharides from nutritional and bioactive perspectives. This review aims to provide new resources and accumulate data for the research and development of natural functional food ingredients and to provide a new theoretical basis for the further development and utilization of hawthorn pectin and its derived oligosaccharides.

## 2. Extraction Methods

The extraction methods of hawthorn pectin mainly include solvent extraction, physical extraction, enzymatic extraction, and combined extraction (Table 1). Hawthorn pectin extracted by different methods displays different physicochemical indexes, such as monosaccharide composition and molecular weight, thus resulting in diverse functional properties [25]. The results showed that the enzyme-assisted hot water extraction method gave higher extraction rate, viscosity, and degree of esterification than the conventional water extraction. Furthermore, ultrasonic- and microwave-assisted extraction is simple, efficient, and safe. Importantly, combined extraction will be a key research and application area in the future. The combined extraction method has the advantages of high efficiency, extraction rate, and purity, which constitutes an effective way to obtain hawthorn pectin and its derived oligosaccharides. Specifically, the enzyme–ultrasound-assisted extraction method exhibited a higher extraction yield than the single-extraction method. Pectin oligosaccharides from hawthorn (HPOS), small molecular polymers, are mainly prepared from hawthorn pectin by enzymolysis. In this section, the different extraction methods of hawthorn pectin and HPOS are briefly introduced.

### 2.1. Solvent Extraction Method

Solvent extraction method for hawthorn pectin mainly involves water extraction and acid extraction. The preparation of hawthorn pectin berries by hot water immersion is one of the more traditional extraction methods, which is easy to operate and gentle [26,27]. It was indicated that the extraction yield by hot water with direct stirring extraction was up to 20.5% [23]. The physicochemical properties of hawthorn pectin prepared by freeze-drying-assisted hot water leaching were better than those treated by hot air drying [28,29]. Low-methoxyl homogalacturonic acid-type hawthorn pectin using a continuous hot water leaching method was prepared, with a proportion of homogalacturonic acid structural domains of 81.13% and molecular weight of 348.43 kDa [30]. According to a report by Jiang et al., it was shown that two types of high-methoxylated pectin from fermented and soaked hawthorn dregs using hot water extraction were obtained [31]. Among them, the proportions of GalA were 83.2% and 90.3%, respectively. Pectin, prepared by hot water extraction and column elution of diethylaminoethyl cellulose (DEAE), was rich in rhamnogalacturonic acid [32]. In a previous paper, three kinds of pectin from freeze-dried hawthorn by hot water extraction and gradient ethanol precipitation were prepared [33]. The results showed that the content and esterification rate of GalA gradually decreased with the increase in ethanol concentration. Water extraction and alcohol precipitation is a traditional method for extracting hawthorn pectin. It has the strength of low costs and high safety, but the extraction yield is slightly lower.

Some polysaccharides containing acidic groups, such as hawthorn pectin, are not easily dissolved under acidic conditions and can be treated with acid and then precipitated with ethanol or insoluble complexes. Previous results suggested that the extraction yield of hawthorn pectin was up to 20.9% by the citric acid method [34]. In addition, another study suggested that the highest extraction yield of hawthorn pectin was achieved at pH 2.76/9 °C/citric acid for 120 min [35]. Four types of low-methoxyl pectin were prepared from dried hawthorn using citric acid extraction, all of which were rich in GalA but with different monosaccharide compositions [36]. Similarly, two types of low-methoxyl pectin were extracted from hawthorn wine pomace using hydrochloric acid and citric acid with GalA percentages of 67.21% and 72.24%, respectively [37]. The results indicated that different acid extraction methods had significantly diverse effects on the structure and physicochemical properties of hawthorn pectin. Nevertheless, no significant difference in the extraction rate of hawthorn pectin with different kinds of acids (hydrochloric, sulfuric, nitric, and phosphoric acids) was found [38]. Intriguingly, the yields of 14 varieties of pectin from hawthorn extracted by hydrochloric acid were different [39]. Regarding the acid extraction, it showed a higher yield of pectin than water extraction. Unfortunately, acid extraction often introduces impurities, which might very easily destroy the three-dimensional structure and biological activity of polysaccharides and bring difficulties to the subsequent processes. In recent years, deep eutectic solvents have been gradually used in pectin extraction. Under the optimal conditions, the yield of hawthorn pectin was 4.33% when the molar ratio of choline chloride to urea was 1:3, the liquid/feed ratio was 30:1 (mL/g), the extraction temperature was 80 °C, and the extraction time was 60 min [40].

### 2.2. Physical Extraction Method

Ultrasound, as a physical crushing technique, can efficiently avoid the effect of high temperature on extracts. It has been well recognized for its wide range of advantages, such as simplicity, high efficiency, and solvent saving, and it is widely used in the extraction of hawthorn pectin. Ultrasound-assisted extraction of hawthorn pectin leads to a high yield with low molecular weight, viscosity, and degree of esterification [41]. It is worth noting that ultrasonic-assisted extraction of hawthorn pectin displayed a high content of total sugar and polygalacturonic acid [26,42]. Under the conditions of solid–liquid ratio of 1:15, ultrasonic power of 425 W, and ultrasonic time of 55 min, the extraction rate of hawthorn pectin was 6.15% [43].

Microwaves can rupture the cells through the penetration of high-frequency electromagnetic waves and improve the efficiency of destroying the plant cell wall, thus releasing more pectin. It has been reported that pectin was extracted from dry hawthorn through microwave method [38,44]. Coincidentally, another recent study extracted hawthorn pectin by microwave method, and the percentage of GalA was 58.32% [45]. Under the conditions of sodium hexametaphosphate dosage of 1.35%, solid–liquid ratio of 1:9, microwave power of 440 W, and microwave time of 80 s, the extraction rate was 72.89% [37]. Although ultrasonic- and microwave-assisted extraction have the advantages of simplicity, efficiency, and safety from contamination, they may cause the extract components to be more complex, resulting in difficult separation.

### 2.3. Enzymatic Hydrolysis Extraction Method

Biological enzymes are used to degrade the macromolecules of hawthorn and then release the pectin. It was found that polygalacturonase can effectively hydrolyze pectin from hawthorn [46]. Polygalacturonase, also known as “acid pectinase”, can specifically cut the glycosidic bond between the two GalA units that have not undergone methyl esterification and then hydrolyze the long-chain pectin into small molecules with high efficiency. Hou et al. used xylanase from Trichoderma viride to extract hawthorn pectin with a yield of 17.7% and a high degree of esterification [47]. Compared with physical and chemical methods, the enzyme method has many advantages, including mild reaction conditions, lower cost, and environmental friendliness, and it can achieve industrial production.

HPOS was prepared by enzymatic hydrolysis of hawthorn pectin. Li et al. prepared hawthorn pectin by hot water extraction and treated it with pectinase from *Aspergillus niger* to obtain low molecular hydrolysate (200 Da < MW < 6000 Da) fraction, namely HPOS [48]. Endo-polygalacturonase containing pectinesterase from *Aspergillus niger* was used to cleave hawthorn pectin, and the results showed that ultrafiltered hawthorn pectin hydrolysates yielded 10 fractions of oligosaccharides with degrees of polymerization of 2–11, linked by α–(1→4) linkages [26]. Based on this important research, a penta-oligogalacturonide derived from hawthorn pectin (HPPS) was prepared using column chromatography of DEAE-Sephadex A-25 [49,50,51].

### 2.4. Combined Extraction Method

In order to increase extraction yield or obtain high purity and low molecular weight of hawthorn pectin, enzyme, physical, and chemical methods are often used in combination. A new type of low-molecular-weight pectin from hawthorn was obtained by successive citric acid extraction, pectinase modification, and gradient ethanol precipitation [52]. In another study, pectin was extracted from hawthorn pulp by hydrochloric acid extraction along with microwave [53]. Fresh hawthorn was used as raw material to extract pectin using ultrasound combined with pectinesterase, and the extraction rate was 25.84% [54]. Quaternized hawthorn pectin derivatives with different degrees of substitution were obtained by ultrasound sodium citrate-assisted extraction [55]. High-pressure treatment (HPP) has been reported to cause a reduction in the molecular weight of pectin from potato peel [56]. As reported by Tian et al., hawthorn pectin from cloudy hawthorn juice was extracted by acid heating and HPP [57]. The extraction rate of pectin ranged from 37.40% to 40.26%, with the highest extraction rate at 600 MPa/6 min treatment. Thus, we infer that enzyme-HPP-assisted extraction or enzyme–microwave-assisted extraction methods may also be an effective choice to improve the yield of hawthorn pectin based on the desired outcome. Ultrasonic-assisted enzymatic degradation of hawthorn pectin was carried out to obtain HPOS, according to Zhu et al. [58]. After fractionation, the yields of the low-molecular-weight HPOS (MW < 700 Da), medium-molecular-weight HPOS (700 Da < MW < 3000 Da), and high-molecular-weight HPOS (MW > 3000 Da) were 12.6%, 80.9%, and 6.5%, respectively.

**Table 1 foods-13-02750-t001:** Extraction methods and physicochemical indexes of hawthorn pectin and its derived oligosaccharides.

Methods	Principles	Materials	Extraction Conditions	Extraction Rate	Physicochemical Indexes	References
Hot water extraction	Effectively disrupts plant cell walls and promotes chemical component release	hawthorn pectin	solid–liquid ratio of 1:10, temperature of 95 °C, time of 4 h	8.7%	total sugar 91.7%, GalA 63.4%, protein 4.9%, methoxyl 2.5%	[23,27]
		hawthorn pectin	solid–liquid ratio of 1:20, temperature of 90 °C, time of 3 h	10.1%~13.7%	total sugar 47.8~76.5%, GalA 41.5~60.1%, DE 81.5~83.1%	[28]
		hawthorn pectin	solid–liquid ratio of 1:20, temperature of 90 °C, time of 2 h	9.14%~14.48%	protein 2.56~2.69%, GalA 69.1~93.3%, DE 30.9~76.7%	[29,31]
		hawthorn pectin	solid–liquid ratio of 1:10, temperature of 90 °C, time of 4 h	-	GalA 81.72%, DE 29.34%	[30]
		hawthorn pectin	solid–liquid ratio of 1:20, temperature of 70 °C, time of 3 h	-	GalA 28.95%, DE 32.52%	[32]
		hawthorn pectin	solid–liquid ratio of 1:15, temperature of 90 °C, time of 2 h	0.11%~3.29%	protein 1.27~1.68%, GalA 31.83~73.09%, DE 23.08~62.21%	[33]
Acid extraction	Polysaccharides containing acidic groups can be treated with acid and then precipitated with ethanol or insoluble complexes	hawthorn pectin	citric acid (4% *w*/*v*), solid–liquid ratio of 1:10, temperature of 85 °C, time of 2 h	19.8%~20.9%	protein 2.8~3.5%, GalA 86.0~86.7%, DE 78.1~78.2%	[34]
		hawthorn pectin	citric acid/lemon juice (7% *v*/*v*), solid–liquid ratio of 1:10, temperature of 60/90 °C, time of 2 h	16.75%/7.32%	anhydrouronic acid 122.90%/86.21%, DE 54.16%/53.34%, acetyl 0.68%/0.76%	[35]
		hawthorn pectin	citric acid (2 mol/L), solid–liquid ratio of 1:30, temperature of 85 °C, time of 2 h	-	protein 0.73~2.11%, GalA 82.43~90.27%, DE 37.67~39.80%	[36]
		hawthorn pectin	hydrochloric acid/citric acid (2 mol/L), solid–liquid ratio of 1:6, temperature of 85 °C, time of 2 h	61.05%/67.81%	protein 0.11%/0.12%, GalA 67.21%/72.24%, DE 37.95%/25.50%	[37]
		hawthorn pectin	hydrochloric/sulfuric/nitric/phosphoric acids, solid–liquid ratio of 1:40, temperature of 80 °C, time of 80 min	4.25%/4.40%/ 2.40%/3.80%	-	[38]
		hawthorn pectin	sulfuric acid (1 mol/L), solid–liquid ratio of 1:10, temperature of 85 °C, time of 25 min	0.71%~4.11%	total sugar 19.05~50.20%, GalA 51.26~55.47%	[39]
		hawthorn pectin	hydrochloric acid (1.5 mol/L), solid–liquid ratio of 1:15, temperature of 70 °C, time of 40 min	3.07%	GalA 78.06%, DE 63.46%	[40]
Ultrasound extraction	Accelerate the release, diffusion, and dissolution of effective substances in the cell	hawthorn pectin	ultrasound (20 kHz, 130 W), solid–liquid ratio of 1:6, temperature of 85 °C, time of 10 min	-	GalA 70.85~79.91%, DE 13.88~24.66%	[41]
		hawthorn pectin	ultrasound (500 W), solid–liquid ratio of 1:15, temperature of 80 °C, time of 10 min	6.0%	total sugar 85.0%, GalA 75.1%, DE 86.2%	[42]
		hawthorn pectin	ultrasound (425 W), solid–liquid ratio of 1:15, temperature of 74 °C, time of 55 min	6.15%	-	[43]
Microwave extraction	High-frequency electromagnetic waves penetrate the extraction medium, causing the cell to rupture and the active ingredient to dissolve	hawthorn pectin	microwave (440 W), solid–liquid ratio of 1:9, time of 80 s	72.89%	protein 0.03%, GalA 68.94%, DE 31.65%	[37]
		hawthorn pectin	microwave (700 W), solid–liquid ratio of 1:30, time of 120 s	5.7%	-	[38]
		hawthorn pectin	microwave (700 W, 2450 Hz), solid–liquid ratio of 1:124, time of 2.11 min	6.59%	-	[44]
		hawthorn pectin	microwave (440 W), solid–liquid ratio of 1:45, time of 80 s	-	total neutral sugars 4.73%, GalA 58.32%, DE 42.96%	[45]
Enzymatic hydrolysis extraction	Cell wall components were hydrolyzed, causing a reduction in the resistance to mass transfer of active ingredients from the extracellular medium	hawthorn pectin	cellulase (80 U/g), solid–liquid ratio of 1:8, temperature of 60 °C, time of 4 h	62.29%	protein 0.24%, GalA 65.93%, DE 8.84%	[37]
		hawthorn pectin	xylanase (70 U/g), solid–liquid ratio of 1:15, temperature of 50.5 °C, time of 3 h	16.8%	-	[42]
		hawthorn pectin	pectinase (0.9% *w*/*v*), solid–liquid ratio of 1:10, temperature of 48 °C, time of 4 h	-	-	[46]
		pectin oligosacchari-des from hawthorn	endo-polygalacturonase containing pectinesterase from *Aspergillus niger* (0.2 U/mL), temperature of 50 °C, time of 2 h	4.9–13.3%	total sugar 98.4~99.3%, uronic acid 97.1~99.7%, DP 2–11	[26]
		pectin oligosacchari-des from hawthorn	pectinase (0.2 U/mL), temperature of 45 °C, time of 3 h	-	total sugar 99.7%, uronic acid 93.6%, MW 200~6000	[48]
		pectin oligosacchari-des from hawthorn	endo-polygalacturonase (0.2 U/mL), temperature of 50 °C, time of 3 h	11.4%	uronic acid 99.2%, DP 5, molecular weight 898	[49,50,51]
Combined extraction	In order to improve the extraction rate or obtain hawthorn pectin with high purity and low molecular weight, enzyme, physical, and chemical methods are often used in combination	hawthorn pectin	hydrochloric acid (1.5 mol/L); ultrasound, solid–liquid ratio of 1:15, temperature of 70 °C, time of 40 min	3.32%	GalA 83.66%, DE 65.51%	[40]
		hawthorn pectin	hydrochloric acid (1 mol/L); ultrasound (240 W), solid–liquid ratio of 1:40, temperature of 80 °C, time of 1 h	-	total neutral sugars 6.17%, GalA 80.00%, DE 51.96%	[45]
		hawthorn pectin	acetic acid (0.2 mol/L); pectinase (12 U/mL), solid–liquid ratio of 1:100, temperature of 40 °C, time of 4 h	-	GalA 77.81~82.81%	[52]
		hawthorn pectin	hydrochloric acid (0.05 mol/L); microwave (800 W), solid–liquid ratio of 1:15, time of 50 s	5.87%	-	[53]
		hawthorn pectin	ultrasound (288 W); pectinesterase activator (12.28%), solid–liquid ratio of 1:26, temperature of 80 °C, time of 119 min	25.84%	GalA 66.78%, DE 18.34%	[54]
		hawthorn pectin	hawthorn juice (1:7 g/mL), high-pressure (300/600 MPa), temperature of 24/33 °C, time of 2/6 min	37.4%~40.3%	GalA 43.85~45.17%, DE 34.56~39.51%	[57]
		pectin oligosacchari-des from hawthorn	endo-polygalacturonase (0.2 U/mL), temperature of 50 °C, time of 3 h; ultrasound power of 800 W, frequency of 20 kHz, time of 20 min	low, medium, and high molecular weights were 12.6%, 80.9%, and 6.9%, respectively	carbohydrate 99.4~99.7%, uronic acid 56.6~86.3%	[58]

Note: GalA: galacturonic acid; DE: degree of esterification; DP: degree of polymerization.

## 3. Functional Activities

Hawthorn, as a unique medicinal food, has been widely used in functional food and traditional Chinese medicine. Hawthorn is rich in pectin, which ranks high among fruits. At present, numerous studies on the functional activities of hawthorn pectin mostly focus on antioxidants, regulating lipid metabolism, and antimicrobial activities (Figure 2). The beneficial effects of hawthorn pectin and its derived oligosaccharides mentioned above are briefly discussed below (Table 2).

### 3.1. Improvement of Antiglycation and Antioxidant Properties

Non-enzymatic glycosylation is a complex aminocarbonyl reaction between reducing sugars and proteins, in which protein is modified by glucose to form advanced glycation end products (AGEs) [59]. AGEs are one of the major risk factors for many chronic diseases, such as aging, atherosclerosis, and diabetic complications [60]. It has been reported that hawthorn pectin exhibited strong antiglycation and antioxidant properties [37]. A recent study extracted hawthorn polysaccharides, including neutral and acidic polysaccharides [61]. In particular, acidic polysaccharides have higher anti-glycation activity than that of neutral polysaccharides. Generally, ABTS+· radicals, DPPH· radicals, and hydroxyl radicals, as the most common free radicals, may play a crucial role in the evaluation of antioxidant activity [62]. The antioxidant mechanisms of hawthorn pectin involve several pathways and processes that contribute to their ability to counteract oxidative stress (Figure 3). Previous studies have demonstrated that hawthorn pectin was able to scavenge ABTS+·, DPPH·, and -OH radicals [40,63], which makes it an effective free radical scavenger [64]. Representative in vivo results show that hawthorn pectin significantly increased the activities of glutathione peroxidase (GSH-PX), superoxide dismutase (SOD), and catalase (CAT) and reduced the content of malondialdehyde (MDA) in the liver of mice [65]. Moreover, hawthorn pectin can modulate signaling pathways involved in oxidative stress and inflammation, such as the nuclear factor kappa-B (NF-κB)/mitogen-activated protein kinase (MAPK)/nuclear factor-E2-related factor 2 (Nrf2) pathway. The activation of this pathway can result in the downregulation of antioxidant genes and enzymes and provide a cellular defense against oxidative damage [1].

Pectin oligosaccharides can be obtained by enzymatic or ultrasonic hydrolysis of pectin [66,67]. It was reported that HPOS exhibited high antiglycation activity by inhibiting the formation of AGEs [58,59]. HPOS significantly increased the activity of SOD in serum, while the synthesis and accumulation of the oxidative product MDA in serum was significantly inhibited [48]. Intriguingly, HPPS exhibited concentration-dependent scavenging activities against O_2_-·, -OH, and DPPH· radicals. HPPS administration also significantly increased SOD, GSH-PX, and CAT enzyme activities but lowered the MDA content in the liver of high-fat diet (HFD)-fed mice [13].

### 3.2. Regulation of Lipid Metabolism

HFD disrupts the balance of lipid metabolism in the organism, triggering a series of adverse reactions, including endocrine disruption, insulin resistance, oxidative stress, and the release of inflammatory factors, which further induces the development of obesity and related complications [68,69]. Compared with drug administration, proper diet and nutrition management are more acceptable for consumers to reduce the rate of obesity-associated diseases. Hawthorn pectin has good inhibitory effect on lipid digestion due to its large molecular weight and highly branched chains [70]. It was found that hawthorn pectin could inhibit hypercholesterolemia, ameliorate lipid accumulation in the liver, and enhance the excretion of bile acid in HFD hamsters [71]. According to the report by Zhang et al., this may be due to the fact that cholesterol micelles can be wrapped around the main and side chains of hawthorn pectin to form complexes and then accelerate cholesterol consumption [72]. Furthermore, hawthorn pectin could promote the excretion of cholesterol by restoring the imbalance of gut microbiota [73].

The beneficial effects of degradation products from hawthorn pectin on lipid metabolism have been intensively studied in recent years. In general, HPOS could exert healthy functions on lipid metabolism through the inhibition of fatty acid synthesis and nuclear factor kappa-B (NF-κB) activation, promotion of adiponectin synthesis, and regulation of the AMP-activated protein kinase (AMPK) signaling pathway (Figure 4). It is worth noting that HPOS effectively reduced the serum total cholesterol (TC) and low-density lipoprotein cholesterol (LDL-C) levels in HFD mice [49]. Meanwhile, HPOS significantly decreased the content of total fat in the liver as well as pro-inflammatory factors, tumor necrosis factor (TNF)-α, and interleukin-6 (IL-6) levels in the liver of HFD mice [50]. Moreover, HPOS could ameliorate hepatic lipid oxidation by promoting adiponectin synthesis and activation [51]. The influence of HPOS on the adiponectin signaling pathway and white adipose metabolism in HFD mice was investigated. The results demonstrated that HPOS could improve lipid metabolism of adipose tissue through activating adiponectin receptor (AdipoR1)/AMPK/peroxisome proliferator-activated receptor alpha (PPARα) signaling pathway [74]. Moreover, HPPS inhibiting fatty acid synthesis and improving insulin sensitivity in obese mice has been reported [75]. In addition, HPPS restrained the reabsorption of bile acids in the ileum and improved the metabolism of cholesterol. It was speculated that HPPS was a competitive inhibitor of sodium-dependent bile acid transporters, thus inhibiting bile acid reabsorption in the ileum and improving cholesterol metabolism [76]. Results of another study suggested that HPPS could be used as a dietary supplement for the prevention of fatty liver and oxidative damage [13]. Meanwhile, Li et al. suggested that the continuous intake of HPPS may be applicable as a dietary therapy to inhibit obesity and cardiovascular diseases [77]. Overall, administration of hawthorn pectin or its degradation product tends to be a good class of lipid-lowering substances with the potential to be developed into lipid-modulating drugs or nutraceuticals.

### 3.3. Presence of Antimicrobial Activity

Among food fibers, pectin is an effective inhibitor of the most widely distributed pathogenic and opportunistic microorganisms [78]. Hawthorn pectin was hydrolyzed to a certain degree by pectinase, and its enzyme degradation products could also show stronger antimicrobial activity [23]. With the increase in hydrolysis time, pectinase will randomly cut polygalacturonic acid into small fragments. Among them, HPOS with an average polymerization degree of 3 had strong inhibition effects on *Escherichia coli*, *Bacillus subtilis*, and *Staphylococcus aureus* [79]. It is hypothesized that HPOS may be capable of disrupting the permeability and integrity of cell membranes, leading to leakage of cellular contents. This may adversely affect metabolic activity, thereby inhibiting bacterial growth. Li et al. found that HPOS had a stronger antibacterial effect on the growth of *Bacillus subtilis* after forming a complex with sodium lactate [80]. Interestingly, HPOS significantly maintained the quality characteristics of shiitake mushrooms by preventing the growth of mold [81]. Consequently, HPOS can be applied as a natural preservative to prevent the growth of spoilage and pathogenic bacteria in food and extend the shelf life of food.

### 3.4. Other Functions

Dietary fiber cannot be digested or absorbed by the small intestine of the human body, which contributes to modulating the health of the intestinal tract, thus promoting defecation and improving constipation [82,83]. Previous studies have confirmed the ability of hawthorn pectin to improve intestinal function, as depicted in Figure 5. The dietary fiber in hawthorn is mainly pectin, which is capable of promoting the rate of small intestinal propulsion, shortening the time to the first black stool, and increasing the volume of defecation in constipated mice [84]. Moreover, hawthorn pectin could ameliorate the gut environment of mice by promoting the growth of probiotics and the accumulation of short-chain fatty acids (SCFAs) [85]. An in vitro fecal fermentation test revealed that hawthorn pectin could enhance the relative abundance of *Bacteroides* and *Faecalibacterium* and was positively correlated with acetic acid and butyric acid [86,87]. However, the potential properties and mechanisms of hawthorn pectin in relation to intestinal modulatory activity are still to be demonstrated in vivo.

Hawthorn pectin can eliminate exercise fatigue. It was found that hawthorn pectin can prolong the time of normobaric hypoxia tolerance and increase the time of weight-bearing swimming in mice by promoting glycogen reserves and reducing lactic acid in the body [88]. Polysaccharides, as important active ingredients, exhibit immunomodulatory activity [89]. According to a report by Luo et al., it was shown that hawthorn polysaccharides had enhanced cellular immunity function [90]. It could be due to its ability to stimulate immune organs, enhance phagocytic cell function, and promote lymphocyte transformation and antibody production, thereby exerting immune regulatory effects [91]. Meanwhile, hawthorn polysaccharides significantly inhibited the proliferation of colon cancer cells by induction of cell cycle block [92]. This was achieved through the regulation of the phosphatidylinositide 3-kinases (PI3K)/protein kinase B (AKT)/mammalian target of rapamycin (mTOR) signaling pathway and further activation of the p38 mitogen-activated protein kinase (P38) signaling pathway, thereby exhibiting potent anti-cancer properties [93]. HPOS is also protective against ultraviolet B (UVB)-induced oxidative damage and photoaging [94].

**Table 2 foods-13-02750-t002:** Functional properties of hawthorn pectin and its derived oligosaccharides in vitro and in vivo.

Functional Properties	Materials	Testing Subjects	Result/Mechanism	References
Antiglycation and antioxidant properties	hawthorn pectin	in vitro	scavenge radicals and increase inhibition rate of glycosylation	[37,61,63,64]
	hawthorn pectin	in vivo, 50/150/300 mg/kg·bw for 10 weeks, Kunming mice	increase the activities of antioxidant enzymes, as well as reduce content of malondialdehyde in liver	[65]
	pectin oligosaccharides from hawthorn	in vitro; in vivo, 50/150/300 mg/kg·bw for 6 weeks, Kunming mice	scavenge radicals and increase antioxidant enzyme activities	[13]
	pectin oligosaccharides from hawthorn	in vivo, 50/150/300 mg/kg·bw for 6 weeks, Kunming mice	increase antioxidant enzyme activities	[48]
	pectin oligosaccharides from hawthorn	in vitro	inhibit the formation of advanced glycation end-products	[58,59]
Regulation of lipid metabolism	hawthorn pectin	in vitro	possesses a good inhibitory effect on lipid digestion	[70]
	hawthorn pectin	in vitro	wrap cholesterol micelles to form complexes that accelerate the consumption of cholesterol	[72]
	hawthorn pectin	in vivo, 250 mg/kg·bw for 6 weeks, Sprague Dawley rats	enhance the excretion of cholesterol and bile acid and restore the imbalance of intestinal microbiota	[73]
	pectin pentasaccharide from hawthorn	in vivo, 50/150/300 mg/kg·bw, 4/10 weeks, Kunming mice	increase the content of serum high-density lipoprotein cholesterol, fecal bile acids, and the gene expressions of cholesterol 7α-hydroxylase	[49]
	pectin oligosaccharides from hawthorn	in vivo, 0.25/0.75/1.5 g/kg·bw for 10 weeks, Kunming mice	ameliorate hepatic inflammation via NF-κB inactivation	[50]
	pectin oligogalacturonide from hawthorn	in vivo, 0.25/0.75/1.5 g/kg·bw for 10 weeks, Kunming mice	facilitate the synthesis and activation of adiponectin to improve hepatic lipid oxidation	[51]
	hawthorn pectin and its hydrolyzates	in vivo, 300 mg/kg·bw for 4 weeks, hamsters	prevent high-fat diet-induced hypercholesterolemia, improve hepatic lipid accumulation, and promote fecal bile acid excretion	[71]
	pectin oligosaccharides from hawthorn	in vivo, 0.25/0.75/1.5 g/kg·bw for 10 weeks, Kunming mice	activate adiponectin-mediated AdipoR1/AMPK/PPARα signaling path in white adipose tissue	[74]
	pectin pentaglaracturonide from hawthorn	in vivo, 50/150/300 mg/kg·bw, 4/10 weeks, Kunming mice	inhibit fatty acid synthesis and improve insulin sensitivity	[75]
	pectin penta-oligogalacturonide from hawthorn	in vivo, 300 mg/kg·bw for 4 weeks, Kunming mice	suppress intestinal bile acids absorption and downregulate the FXR-FGF15 axis	[76]
	pectin pentaoligosaccharide from hawthorn	in vivo, 150 mg/kg·bw, 10 weeks, Kunming mice	increase the hepatic fatty acid oxidation-related enzyme activities and mRNA levels	[77]
Antimicrobial activity	pectin oligosaccharides from hawthorn	in vitro	inhibit bacterial growth of *Escherichia coli*, *Bacillus subtilis*, and *Staphylococcus aureus*	[79,80]
	pectin oligosaccharides from hawthorn	in vitro	maintain the quality characteristics of shiitake mushrooms by inhibiting the growth of mold	[81]
Improvement of intestinal function	hawthorn pectin	in vivo, 250/500/1000 mg/kg·bw for 40 d, ICR mice	promote the rate of small intestinal propulsion and increase the volume of defecation in constipated mice	[84]
	hawthorn pectin	in vivo, 1/2/4 g/kg·bw for 6 weeks, Kunming mice	promote the growth of probiotic bacteria and accumulation of short-chain fatty acids	[85]
	hawthorn pectin	in vitro	increase the relative abundance of *Bacteroides* and *Faecalibacterium*	[86,87]
Anti-fatigue activity	hawthorn pectin	in vivo, 1/2/4 g/kg·bw for 6 weeks, Kunming mice	prolong the time of normobaric hypoxia tolerance and increase the time of weight-bearing swimming	[88]
Immunomodulatory activity	hawthorn pectin	in vitro	promote the proliferation of spleen lymphocytes	[90]
Anti-cancer activity	hawthorn pectin	in vitro	inhibit proliferation of HCT116 cells	[92]
Anti-photoaging activity	hawthorn pectin	in vitro	protect against oxidative damage and photoaging induced by ultraviolet B	[94]

## 4. Structure–Activity Relationship

The bioactivity of hawthorn pectin and its derived oligosaccharides is closely tied to their structural characteristics. Factors such as monosaccharide composition, average molecular weight, chemical structures, and conformational features play vital roles in determining their bioactivity [95,96]. However, the structure–activity relationship of hawthorn pectin and its derived oligosaccharides has been rarely reported in the existing literature. Some general relationships can be inferred based on recent studies.

Interestingly, the antioxidant effect of hawthorn pectin is closely related to its monosaccharide composition and average molecular weight. Three pectin fractions (CPP-1, CPP-2, and CPP-3) were extracted and purified from hawthorn, which exhibited different monosaccharide compositions, molecular weight characteristics, and levels of uronic acid and sulfate groups. The results demonstrated that CPP-2 with lower molecular weight and higher content of uronic acid and sulfate groups exhibited higher antioxidant capacity. The finding has shown that hawthorn pectin with lower molecular weight and higher uronic acid content generally indicated higher biological activities. Moreover, the previous literature has indicated that a higher content of sulfate groups can enhance the bioactivity of polysaccharides [97,98]. Meanwhile, Shang et al. suggested that the acidic polysaccharides from hawthorn exhibited higher anti-glycation activity than that of neutral polysaccharides [61]. This implied that the high anti-glycosylation activity of hawthorn pectin was related to its high content of GalA and low degree of molecular branching. It seems that hawthorn pectin, with a lower molecular weight and higher contents of uronic acid, GalA, and sulfate group, tends to exhibit stronger bioactivities.

Additionally, the mode of glycosidic bond linkage, the location of branch chains, and the sequence of main and side chains can also influence the biological activity of hawthorn pectin [1]. Previous studies indicated that the main modes of glycosidic bond linkage in polysaccharides include α-1,4, β-1,3, and β-1,6, and different linkage modes were associated with various bioactivities, such as antioxidant activity, antitumor activity, immunomodulatory activity, etc. [99,100]. Therefore, it is crucial to explore their impact on the bioactivities of hawthorn pectin through further investigation of the structural characteristics.

Similarly, the antiglycation activity of HPOS was correlated with its monosaccharide composition and molecular weight. Previous studies suggested that low-molecular-weight HPOS (GalA accounts for 90.1%) has almost no antiglycation activity, which may be due to the possibility that a large amount of monosaccharide is contained [58]. These monosaccharides may not only have no antiglycation activity but also may promote the non-enzymatic glycation reaction [101]. Interestingly, the antiglycation activity of medium-molecular-weight HPOS was significantly stronger than that of high-molecular-weight HPOS. The cause of these differences in antiglycation activity may be related to their monosaccharide composition and molecular weight.

## 5. Processing and Application

As a natural compound, pectin has been extensively used in foodstuffs [17]. The unique chemical properties and structure of hawthorn pectin make it a good emulsifying and gelling property and biocompatibility, which can be applied in the food industry as an emulsifier and gelling agent, etc., and advance the processing of hawthorn products. Some major processing and applications of hawthorn pectin and its derived oligosaccharides are briefly summarized as follows.

### 5.1. Emulsifiers and Gelling Agents

Currently, there is an increasing demand for natural food ingredients and emulsifiers. Pectin extracted from citrus and apple is reportedly not recommended for use as an emulsifier [102]. Recent studies have indicated that hawthorn pectin is characterized by high viscosity and shows better emulsification and stability than that of citrus pectin [34]. In particular, hawthorn pectin obtained by hot air drying had good emulsifying ability, while pectin obtained by blackening displayed poor emulsifying properties [29]. The results indicated that the emulsification performance of high molecular weight was better. In addition, emulsifiers made from substances such as hawthorn pectin can be used as coatings for fresh produce. It was found that emulsifier films made by high-esterified hawthorn pectin were better than citrus pectin [22]. It is worth noting that the development of natural polymer nanoparticles as stabilizers for Pickering emulsion has attracted increasing attention. Hawthorn pectin can be used as a particulate shell material to stabilize Pickering emulsions, protecting the lipid components from oxidation, as demonstrated by Jiang et al. [103]. *Litsea cubeba* oil-loaded Pickering emulsion (LOPE) was prepared with hawthorn pectin/β-cyclodextrin complex particle as a stabilizer, exhibiting excellent physical stability [104].

Hawthorn pectin is prone to gel formation under acidic conditions in the presence of sucrose. Linares-García et al. found that the hardness, gelatinous degree, and chewiness of hawthorn pectin were 10, 31, and 46 times greater than citrus pectin gels, respectively [22]. A study showed that hawthorn pectin was the main substance that influenced the sensory and physical properties of hawthorn yogurt [105]. In conclusion, numerous studies have demonstrated that pectin plays a crucial role in hawthorn gel formation.

### 5.2. Food Packaging Film

Recently, the development of edible packaging materials based on biopolymers has received significant attention [106]. Pectin, due to its non-toxicity, biocompatibility, biodegradability, and various physicochemical properties, is considered a suitable polymer matrix for the preparation of edible films [107]. Four homogalacturonan types of hawthorn pectin were obtained by hydrochloric and citric acid, ultrasound, and microwave treatment, respectively [45]. The results showed that the high content of GalA and molecular weight of pectin were useful for the preparation of pectin films with excellent properties. Hawthorn fragments were used as raw materials to develop quaternized hawthorn pectin/thyme essential oil composite films with improved water solubility and antibacterial activity [108]. A biodegradable food packaging film obtained by adding hawthorn pectin/β-cyclodextrin Pickering emulsions displayed excellent comprehensive properties [104].

### 5.3. Modification

Generally, pectin exhibits a complex and unstable structure, high molecular weight, and poor solubility. It is difficult to be absorbed through the intestinal tract after entering the organism but is instead degraded by microorganisms in the colon, limiting its bioactivity and its application in functional foods [109]. Studies have shown that the modified pectin possesses better functional characteristics and bioactivity than natural pectin [110,111]. Ultrasound technology is expected to modify the structure, physicochemical properties, and biological activities of polysaccharides via cavitation-induced degradation and depolymerization. After ultrasound modification, the molecular weight of hawthorn pectin was decreased, while the antioxidant activity was significantly increased [40]. Similarly, hawthorn pectin was modified by ultrasound-assisted pectin methylesterase modification, which exhibited stronger antioxidant activity [112]. Zhu et al. found that the use of ultrasound-assisted vitamin modification could promote the oxidative degradation of hawthorn pectin, resulting in a content of reducing sugars greater than 50% [113]. Therefore, adopting a variety of methods to modify hawthorn pectin can enable it to perform a wider range of functional properties, thus expanding the application scope of hawthorn pectin.

### 5.4. Current Products

#### 5.4.1. Traditional and Recent Food Applications

Hawthorn, with its rich pectin and various functional components, has been widely used in the food industry (Figure 6A) [114]. This is mainly owing to the fact that with the addition of sucrose and the reduction of water molecules, the pectin molecules come closer to each other and gradually form long-chain micelles, eventually aggregating to form a relaxed three-dimensional network structure (Figure 6B). Initially, original leisure food emerged, including sugar-coated haws, hawthorn preserves, hawflakes, hawthorn leathers, fruit jam, etc. [115,116,117]. Dried hawthorn products were developed, including crispy fruit and candy [118]. The emergence of hawthorn juice and hawthorn wine occurred in the 1980s and 1990s [119,120,121]. Subsequently, new types of hawthorn products have been developed, of which the more popular ones include hawthorn yogurt, hawthorn jelly, and hawthorn drink. In the food industry, hawthorn pectin can serve as a natural sweetener in food, providing a healthier alternative to refined sugar or artificial sweeteners because of its mild and naturally sweet taste [122]. Furthermore, hawthorn pectin and its derived oligosaccharides can contribute to the improvement of the body’s health. Thus, hawthorn pectin and its derived oligosaccharides have the potential to be used as a natural, functional food additive. Raposo et al. suggested that hawthorn-derived food supplements displayed beneficial effects on microcirculation [123]. Recently, hawthorn pectin and its derived oligosaccharides have been incorporated into various novel food products, including beverages, snacks, and supplements, with the aim to enhance their properties and potential health advantages. For instance, hawthorn pectin was added to yogurt, which showed better water-holding capacity, stability, and suspension stability by lowering the pH value of yogurt after fermentation [22,124]. Furthermore, hawthorn jelly presented the typical flavor of hawthorn, along with being crystal clear, having sweet and sour tastes, and being elastic and nutritious [125]. The mixed beverage of hawthorn and green tea is jacinth, which has a delicate and uniform tissue and a unique hawthorn and green tea aroma [126]. Similarly, hawthorn berry ultrafine powder and milk were used as the main ingredients, together with pectin oligogalacturonic acid (PGA), to produce hawthorn PGA yogurt drinks [127]. Due to extraordinary antimicrobial activity, HPOS can be used in the shiitake mushroom preservation industry to enhance their physical and functional properties [81]. Consequently, hawthorn pectin and its derived oligosaccharides have been shown to be an important ingredient in the preparation of novel foods.

#### 5.4.2. Nutraceutical and Medicinal Applications

In recent years, hawthorn has also been used in the development of nutraceuticals and medicines due to its functional research and mining. Hawthorn is extensively mentioned in Chinese classical prescriptions, which have been shown to strengthen the spleen and stomach as well as treat conditions such as postpartum abdominal cramping, blood stasis, and constipation [128]. In addition, hawthorn pectin displays the potential as a drug delivery system targeting specific tissues or organs in the body. This is mainly owing to its unique physicochemical properties, such as high viscosity and good biocompatibility, allowing it to encapsulate and deliver bioactive compounds. At present, binary hydrogels composed of hawthorn pectin and protein, with good textural properties and high load efficiency, have become the main research focus in the construction of delivery systems with bioactive compounds (Figure 6C) [129,130,131]. Specifically, zein and sodium alginate are widely used for compositing with hawthorn pectin to form nanoparticles or hydrogel for drug delivery [132,133].

## 6. Conclusions and Perspectives

As a natural compound, hawthorn pectin and its derived oligosaccharides have garnered significant attention and held immense research potential. Numerous methods, mainly hot water extraction, ultrasound- and microwave-assisted extraction, and enzymatic extraction, have been developed, and enzymatic hydrolysis is preferred because of its high efficiency and well-defined composition. Owing to the decrease in molecular weight and the increase in the active group, hawthorn pectin-derived oligosaccharides have better bioactivities, such as antioxidant activity, regulation of lipid metabolism, and antimicrobial activity, than hawthorn pectin itself. The bioactivities of hawthorn pectin-derived oligosaccharides mainly contributed to its monosaccharide composition and molecular weight. Due to the physicochemical characteristics and multiple health benefits, hawthorn pectin and its derived oligosaccharides have been used as functional food materials or incorporated into food matrices to improve the process and organoleptic properties of foods.

In view of the increasing market demand, the following issues should be paid attention to concerning hawthorn pectin and its derived oligosaccharides in the future. Firstly, the application of hawthorn pectin and its derived oligosaccharides in the field of food and medicine is still in the initial stage. The precise chemical structure of hawthorn pectin and its derived oligosaccharides should be further studied to understand the influence of different structural features on its different biological activities. The potential properties of hawthorn pectin, including anti-aging, anti-cancer, and immunomodulatory activities, still need to be demonstrated in vivo, and its roles, which are attributable to pectin itself or to other compounds, still need to be verified. In addition, the widespread application of hawthorn pectin and its derived oligosaccharides in the food and pharmaceutical industries requires the establishment of regulatory approvals and safety assessments. Food and drug supervision departments regularly inspect the quality of the products produced by enterprises. Samples prepared in the laboratory should be used in research. Although no apparent toxicity was observed in rodent studies, a comprehensive evaluation of long-term in vivo toxicity is imperative to address existing knowledge gaps and ensure the safety of hawthorn pectin and its derived oligosaccharides. This review will pave the way for the further development of hawthorn pectin and hawthorn pectin-derived oligosaccharides in functional food and other industries.

## Figures and Tables

**Figure 1 foods-13-02750-f001:**
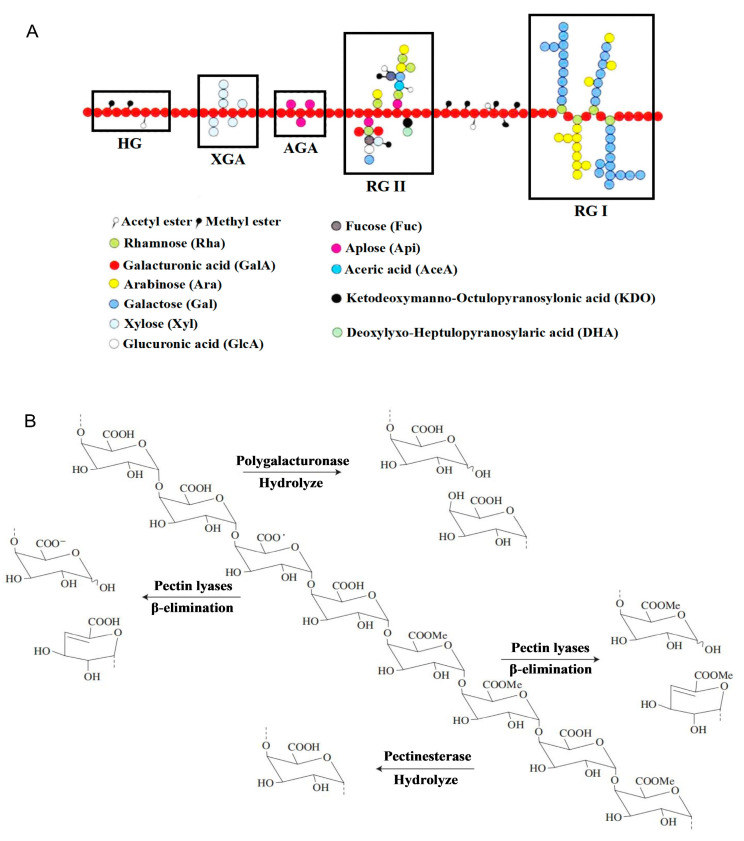
Structure and enzymatic hydrolysis process of pectin. (**A**) Diagram of primary structure of pectin; (**B**) degradation pathways of pectin by pectolytic enzymes.

**Figure 2 foods-13-02750-f002:**
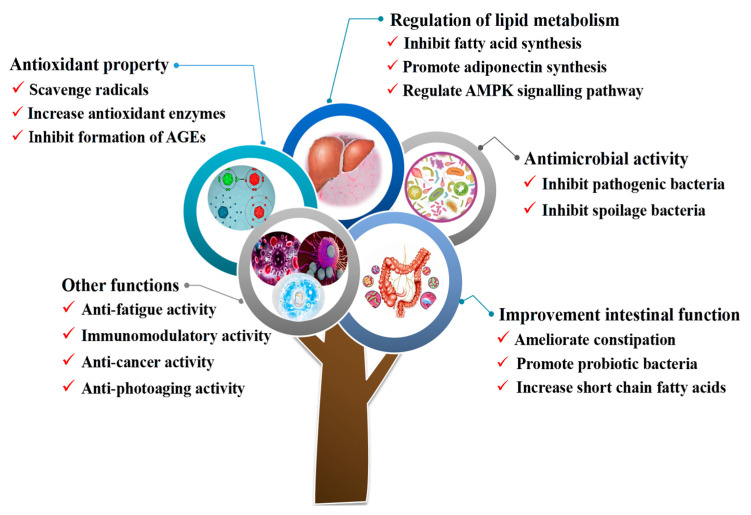
Summary of the biological activities of hawthorn pectin and its derived oligosaccharides in vitro and in vivo.

**Figure 3 foods-13-02750-f003:**
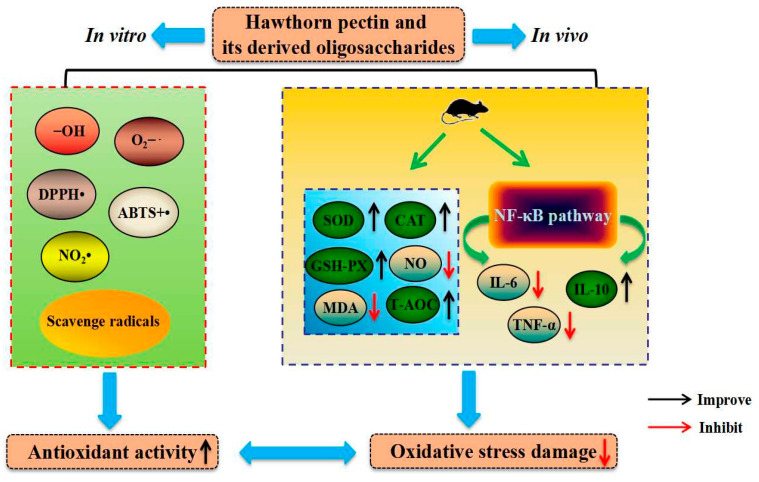
The potential mechanisms of antioxidant activity of hawthorn pectin and its derived oligosaccharides.

**Figure 4 foods-13-02750-f004:**
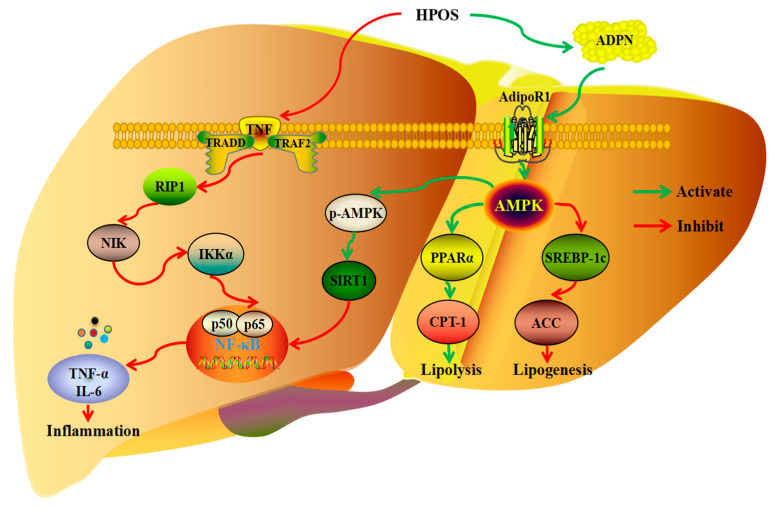
Pectin oligosaccharides from hawthorn (HPOS) regulation pathway of lipid metabolism in liver of mice. Tumor necrosis factor (TNF), nuclear factor kappa-B (NF-κB), receptor-interacting protein kinase 1 (RIP1), tumor necrosis factor (TNF-α), interleukin-6 (IL-6), NF-κB-inducing kinase (NIK), IκB kinase-α (IKKα), tumor necrosis factor receptor-associated death domain protein (TRADD), TNF-α receptor-associated factor 2 (TRAF2), adiponectin (ADPN), adiponectin receptor (AdipoR1), AMP-activated protein kinase (AMPK), and silent information regulator T1 (SIRT1) were labeled with different colors.

**Figure 5 foods-13-02750-f005:**
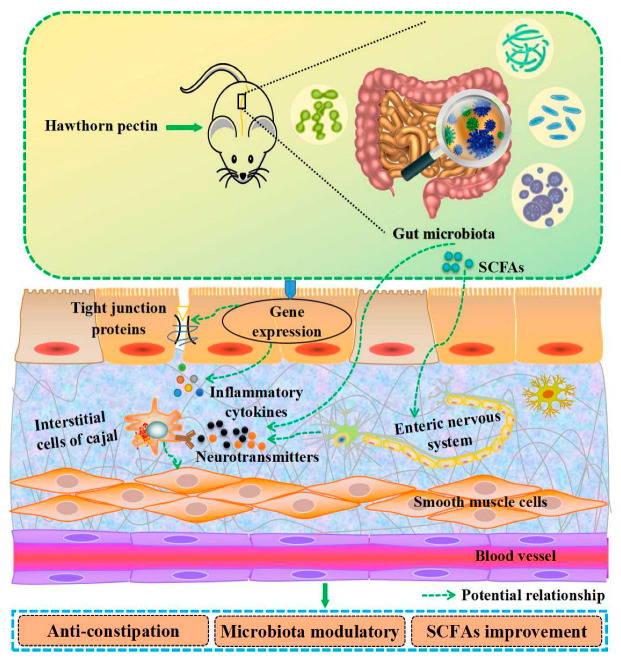
The potential mechanisms of intestinal modulatory activity of hawthorn pectin.

**Figure 6 foods-13-02750-f006:**
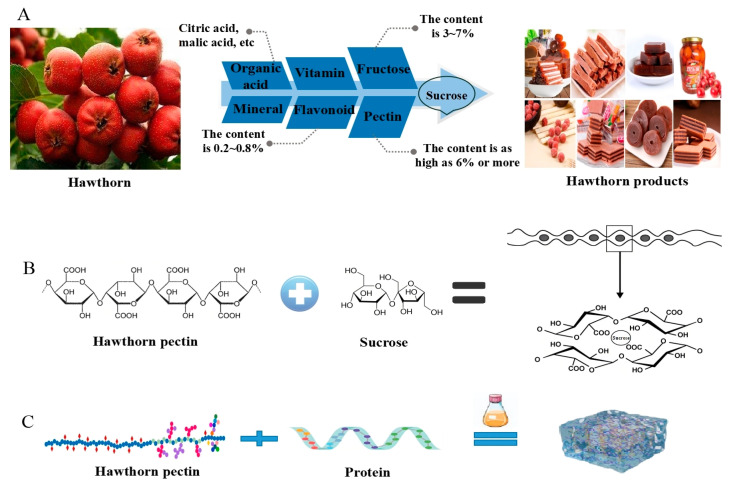
Hawthorn pectin products and their formation mechanism. (**A**) Nutrients in hawthorn and its different market products; (**B**) schematic diagram of possible mechanism in the formation of hawthorn pectin products; (**C**) targeted delivery of hawthorn pectin-protein complexes.

## Data Availability

No new data were created or analyzed in this study. Data sharing is not applicable to this article.

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
