# Peer review of "A Comprehensive Review on the Isolation, Bioactivities, and Structure–Activity Relationship of Hawthorn Pectin and Its Derived Oligosaccharides"

_foods, 2024, doi:10.3390/foods13172750_

Round 1
Reviewer 1 Report
Comments and Suggestions for Authors
The review covers an extensive range of topics related to the extraction, bioactivities, and potential applications of hawthorn pectin and its derivatives. The article is well-structured and thorough in many areas, but it also has some areas that could benefit from additional exploration or clarification.
What should be added for the paper:
- While the review mentions that the bioactivity of pectin can vary based on factors like maturity, storage period, and extraction process, it could have provided more detail on how these factors influence the final product. A more in-depth analysis of the variability in pectin quality and how it affects bioactivity would have strengthened the review
- Although the review covers the bioactivities of hawthorn pectin, it lacks a deeper mechanistic explanation of how these activities are achieved at the molecular level. More detailed insights into the biochemical pathways and interactions involved would enhance the reader's understanding of pectin’s bioactivities
-The review should also explore the potential for synergistic effects when hawthorn pectin is combined with other bioactive compounds. Discussing potential synergies could open new avenues for research and application.
Author Response
Dear Professor:
Your comments were much appreciated. We have revised our manuscript (foods-3172260) according to the comments. The following are our point-to-point responses to the comments (Bold-face). Revisions made in the manuscript are marked in red. We hope that the responses and the revised manuscript will be sufficient to make our manuscript suitable for publication in Foods.
Comments 1: While the review mentions that the bioactivity of pectin can vary based on factors like maturity, storage period, and extraction process, it could have provided more detail on how these factors influence the final product. A more in-depth analysis of the variability in pectin quality and how it affects bioactivity would have strengthened the review.
Response: Thanks for the valuable comments. We have supplemented more detail on how these factors influence the final product, and the analysis of the variability in pectin quality and how it affects bioactivity. The following contents have been added in the revised version.
“Usually, pectin rises in the fifth week of fruit setting until the stone is formed, and then the pectin content decreases, leading to the fruit softening because of enzymatic degradation and solubilization of protopectin [18]. The availability of pectin in fruit increases with the increase of storage time [19].”
“Hawthorn pectin extracted by different methods exhibits different physicochemical indexes such as monosaccharide composition and molecular weight, thus resulting in diverse functional properties [25].”
- Maldonado-Celis, M.E.; Yahia, E.M.; Bedoya, R.; Landázuri, P.; Loango, N.; Aguillón, J.; Restrepo, B.; Ospina, J.C.G. Chemical composition of Mango (Mangifera indicaL.) fruit: Nutritional and phytochemical compounds. Front. Plant Sci. 2019, 10, 1073.
- 19.Kumari, N.; Kumar, M.; Radha,Rais, N.; Puri, S.; Sharma, K.; Natta, S.; Dhumal, S.; Damale, R.D.; Kumar, S.; Senapathy, M.; Deshmukh, S.V.; Anitha, T.; Prabhu, T.; Shenbagavalli, S.; Balamurugan, V.; Lorenzo, J.M.; Kennedy, J.F. Exploring apple pectic polysaccharides: Extraction, characterization, and biological activities-A comprehensive review. Int. J. Biol. Macromol. 2024, 255, 128011.
Comments 2: Although the review covers the bioactivities of hawthorn pectin, it lacks a deeper mechanistic explanation of how these activities are achieved at the molecular level. More detailed insights into the biochemical pathways and interactions involved would enhance the reader's understanding of pectin’s bioactivities.
Response: Thanks a lot for the good suggestion. We have provided more detail on the mechanistic explanation of how these activities are achieved at the molecular level. The following contents have been added in the revised version.
“Besides, hawthorn pectin can modulate signaling pathways involved in oxidative stress and inflammation, such as the nuclear factor kappa-B (NF-κB)/mitogen-activated protein kinase (MAPK)/nuclear factor-E2-related factor 2 (Nrf2) pathway. The activation of this pathway can result in the downregulation of antioxidant genes and enzymes, and provide a cellular defense against oxidative damage [1].”
“It could be due to its ability to stimulate immune organs, enhance phagocytic cell function, promote lymphocyte transformation and antibody production, thereby exerting immune regulatory effects [91].”
“This was achieved through the regulation of the phosphatidylinositide 3-kinases (PI3K)/protein kinase B (AKT)/mammalian target of rapamycin (mTOR) signaling pathway and further activation of the p38 mitogen-activated protein kinase (P38) signaling pathway, thereby exhibiting potent anti-cancer properties [93].”
- Li, Y.M.;Zhong, R.F.; Chen, J.; Luo, Z.G. Structural characterization, anticancer, hypoglycemia and immune activities of polysaccharides from Russula virescens. J. Biol. Macromol. 2021, 184, 380–392.
- Hayden, M.S.;Sankar, G. Shared principles in NF-kappa B signaling. Cell 2008, 132(3), 344–362.
Comments 3: The review should also explore the potential for synergistic effects when hawthorn pectin is combined with other bioactive compounds. Discussing potential synergies could open new avenues for research and application.
Response: Thanks for the comments. We have provided the content about the potential for synergistic effects when hawthorn pectin is combined with other bioactive compounds. The following contents have been added in the revised version.
“Litsea cubeba oil-loaded Pickering emulsion (LOPE) was prepared with hawthorn pectin/β-cyclodextrin complex particle as a stabilizer, exhibiting excellent physical stability [104].”
“Specially, zein and sodium alginate are widely used for compositing with hawthorn pectin to form nanoparticles or hydrogel for drug delivery [132,133].”
- Yang, Z.K.; Li, M.R.; Li, Z.H.; Li, Y.X.; Shi, J.Y.; Huang, X.W.; Sun, Y.; Zhai, X.D.; Zou, X.B.; Xiao, J.B. Incorporation of hawthorn pectin/β-cyclodextrin-stabilized Pickering emulsion and Titanium dioxide nanoparticles for improving the physical, biological, and release properties of guar gum/agar/sodium alginate-based bilayer films. Crop. Prod. 2024, 212, 118302.
- Huang, X.; Li, T.P.; Li, S.H. Encapsulation of vitexin-rhamnoside based on zein/pectin nanoparticles improved its stability and bioavailability. Res. Food Sci. 2023, 6, 100419.
- 133.Li, Z.X.; Geng, Y.X.;Bu, K.X.; Chen, Z.T.; Xu, K.; Zhu, C.H. Construction of a pectin/sodium alginate composite hydrogel delivery system for improving the bioaccessibility of phycocyanin. J. Biol. Macromol. 2024, 269, 131969.
Thank you again for reviewing and considering our manuscript for publication. We look forward to hearing from you soon.
With best regards
Sincerely yours
Lei Chen, Ph. D

Reviewer 2 Report
Comments and Suggestions for Authors
Thanks for the document “A Comprehensive Review on Isolation, Bioactivities and Structure-Activity Relationship of Hawthorn Pectin and Its Derived Oligosaccharides”. I considered this is a document well written. I recommend making some improvements.
Line 51. d-galactopyranuronic. Check this word, I think it should be D-galacturonic acid.
Some words should indicate their meaning the first time or put a glossary at the beginning of the document.
Page 14, line 1 and 19. What is HFD and AMPK?
Page 5, line 178 and 180. Use italics “Aspergillus niger”
Page 17, line 112-113 the information was mentioned before (line 51) “The main chain of hawthorn pectin primarily consists of d-galactopyranuronic acid linked by α-1, 4-glycosidic bonds”
Comments on the Quality of English Language
Minor editing of English language required.
Author Response
Dear Professor:
Your comments were much appreciated. We have revised our manuscript (foods-3172260) according to the comments. The following are our point-to-point responses to the comments (Bold-face). Revisions made in the manuscript are marked in red. We hope that the responses and the revised manuscript will be sufficient to make our manuscript suitable for publication in Foods.
Comments 1: Line 51. d-galactopyranuronic. Check this word, I think it should be D-galacturonic acid.
Response: Thanks a lot for the good suggestion. We have revised as “D-galacturonic acid”.
Comments 2: Some words should indicate their meaning the first time or put a glossary at the beginning of the document.
Response: Thanks for the comments. We have revised the words including diethylaminoethyl cellulose (DEAE), AMP activated protein kinase (AMPK), adiponectin receptor (AdipoR1), and peroxisome proliferators-activated receptors alpha (PPARα), which indicated their meaning in the first time.
Comments 3: Page 14, line 1 and 19. What is HFD and AMPK?
Response: Indeed, HFD stands for high-fat diet, which first appeared on page 11, line 64. AMPK stands for AMP-activated protein kinase. We have indicated the meaning of AMPK in the first time.
Comments 4: Page 5, line 178 and 180. Use italics “Aspergillus niger”.
Response: According to the suggestion, we have revised as “Aspergillus niger” in italics.
Comments 5: Page 17, line 112-113 the information was mentioned before (line 51) “The main chain of hawthorn pectin primarily consists of d-galactopyranuronic acid linked by α-1, 4-glycosidic bonds”.
Response: Thanks a lot for your careful suggestion. We have deleted the repetitive sentence.
Comments 6: Minor editing of English language required.
Response: According to the suggestion, we have revised the English language. Revisions made in the manuscript are marked in red.
Thank you again for reviewing and considering our manuscript for publication. We look forward to hearing from you soon.
With best regards
Sincerely yours
Lei Chen, Ph. D

Reviewer 3 Report
Comments and Suggestions for Authors
Foods.- 3172260
Title: A Comprehensive Review on Isolation, Bioactivities and Structure-Activity Relationship of Hawthorn Pectin and Its Derived Oligosaccharides., By Tao Li, Wenhua Ji, Hongjing Dong, Yingqun Wu, Lanping Guo, Lei Chen, and Xiao Wang
This this paper aims to systematically review recent literature on hawthorn pectin and its derived oligosaccharides, covering the extraction methods and functional characteristics. It also aims to provide new resources and accumulate data for the research and development of natural functional food ingredients, and to provide new theoretical basis for the further development and utilization of hawthorn pectin and its derived oligosaccharides. This review is interesting and overall very well done. It covers various issues as the Hawthorn pectin solvent extraction methods, the enzymatic hydrolysis extraction and mixed methods, the functional activities, the antiglycation and antioxidant properties, the regulation of lipid metabolism, the presence of antimicrobial activity, the structure-activity relationship, several functions, and so on and so forth. The English is clear and well written. However, I would advise the authors to pay attention to the reference in lines 295-296, “Production pectin oligosaccharides, J. Biosci. 2019,129(1),16-72”. The title and names of the authors are right?.
Author Response
Dear Professor:
Your comments were much appreciated. We have revised our manuscript (foods-3172260) according to the comments. The following are our point-to-point responses to the comments (Bold-face). Revisions made in the manuscript are marked in red. We hope that the responses and the revised manuscript will be sufficient to make our manuscript suitable for publication in Foods.
Comments 1: However, I would advise the authors to pay attention to the reference in lines 295-296, “Production pectin oligosaccharides, J. Biosci. 2019, 129(1), 16-72”. The title and names of the authors are right?
Response: Thanks a lot for your careful suggestion. We have rechecked the the reference. The title and names of the authors are right. The details are shown in Figure 1.
Thank you again for reviewing and considering our manuscript for publication. We look forward to hearing from you soon.
With best regards
Sincerely yours
Lei Chen, Ph. D

Reviewer 4 Report
Comments and Suggestions for Authors
The paper entitled “A Comprehensive Review on Isolation, Bioactivities and Structure-Activity Relationship of Hawthorn Pectin and Its Derived Oligosaccharides” provides a comprehensive and well-structured review of hawthorn pectin and its derived oligosaccharides. It effectively covers various aspects of the topic, including extraction methods, functional activities, structure-activity relationships, and applications. Despite minor shortcomings presented below, the paper is a valuable contribution to the field of natural product research.
- Abstract: Please change "acquiring many bioactive polymers" to "containing a variety of bioactive polymers."
- In the section on Extraction methods provide more specific recommendations for optimizing extraction conditions based on the desired outcome.
- Suggestion: please consider dividing subsection 5.4 (Current Products) into two subsections: one focusing on traditional and recent food applications, and another focusing on nutraceutical and medicinal applications.
- Conclusion part:
- Please provide more specific examples of the types of in vivo studies that could be conducted.
- Include more details on the specific regulatory requirements that need to be met.
- Please check in vivo and in vitro throughout the whole manuscript. It should be written in italics.
Author Response
Dear Professor:
Your comments were much appreciated. We have revised our manuscript (foods-3172260) according to the comments. The following are our point-to-point responses to the comments (Bold-face). Revisions made in the manuscript are marked in red. We hope that the responses and the revised manuscript will be sufficient to make our manuscript suitable for publication in Foods.
Comments 1: Abstract: Please change "acquiring many bioactive polymers" to "containing a variety of bioactive polymers."
Response: Thanks a lot for the good suggestion. We have revised as “containing a variety of bioactive polymers” in the Abstract.
Comments 2: In the section on Extraction methods provide more specific recommendations for optimizing extraction conditions based on the desired outcome.
Response: Thanks for the comments. We have provided more specific recommendations for optimizing extraction conditions based on the desired outcome in the section on Extraction methods. The following contents have been added in the revised version.
“The combined extraction method has the advantages of high efficiency, extraction rate and purity, which is an effective way to obtain hawthorn pectin and its derived oligosaccharides. Specifically, the enzyme-ultrasound assisted extraction method exhibited the higher extraction yield than single method.”
“Thus, we infer that enzyme-HPP assisted extraction or enzyme-microwave assisted extraction methods may also be an effective choice to improve the yield of hawthorn pectin based on the desired outcome.”
Comments 3: please consider dividing subsection 5.4 (Current Products) into two subsections: one focusing on traditional and recent food applications, and another focusing on nutraceutical and medicinal applications.
Response: Thanks for the comments. We have divided subsection 5.4 (Current Products) into two subsections: 5.4.1. Traditional and Recent Food Applications and 5.4.2. Nutraceutical and Medicinal Applications.
Comments 4: Please provide more specific examples of the types of in vivo studies that could be conducted.
Response: Thanks for the comments. We have provided more specific examples of the types of in vivo studies. The following contents have been added in the revised version.
“The potential properties of hawthorn pectin including anti-aging, anti-cancer and immunomodulatory activity are still to be demonstrated in vivo and its roles which are attributable to pectin itself or to other compounds are also to be verified.”
Comments 5: Include more details on the specific regulatory requirements that need to be met.
Response: Thanks for the comments. We have provided more details on the specific regulatory requirements that need to be met. The following contents have been added in the revised version.
“Food and drug supervision departments regularly inspect the quality of the products produced by the enterprises. Samples prepared in the laboratory should be used in research. Although no apparent toxicity was observed in rodent studies, a comprehensive evaluation of long-term in vivo toxicity is imperative to address existing knowledge gaps and ensure the safety of hawthorn pectin and its derived oligosaccharides.”
Comments 6: Please check in vivo and in vitro throughout the whole manuscript. It should be written in italics.
Response: According to the suggestion, we have revised as “in vivo” and “in vitro” in italics.
Thank you again for reviewing and considering our manuscript for publication. We look forward to hearing from you soon.
With best regards
Sincerely yours
Lei Chen, Ph. D

Reviewer 5 Report
Comments and Suggestions for Authors
The authors present: A Comprehensive Review on Isolation, Bioactivities and Structure-Activity Relationship of Hawthorn Pectin and Its Derived Oligosaccharides
The paper is very well written, covering different extraction methods, with an extensive literature review; functional properties of hawthorn pectin and its derived oligosaccharides in vitro and in vivo; structure-activity relationship, processing and application, conclusions and perspectives.
As you read in the text, the fruit has large amounts of pectin (about 6.4%), in particular according to the literature, I consider that the content is average, there are other fruits richer in pectin content, such as pomegranate, citrus fruits.
The authors should mention the existence of other sources of natural pectins. https://doi.org/10.3390/molecules28227656
Author Response
Dear Professor:
Your comments were much appreciated. We have revised our manuscript (foods-3172260) according to the comments. The following are our point-to-point responses to the comments (Bold-face). Revisions made in the manuscript are marked in red. We hope that the responses and the revised manuscript will be sufficient to make our manuscript suitable for publication in Foods.
Comments 1: As you read in the text, the fruit has large amounts of pectin (about 6.4%), in particular according to the literature, I consider that the content is average, there are other fruits richer in pectin content, such as pomegranate, citrus fruits. The authors should mention the existence of other sources of natural pectins. https://doi.org/10.3390/molecules28227656
Response: Thanks a lot for the good suggestion. We have revised the sentence “Hawthorn fruit has large amounts of pectin (about 6.4%)” as “Hawthorn fruit contains pectin about 6.4%”. Furthermore, we have added the existence of other sources of natural pectins. The following contents have been added in the revised version.
“Natural pectin comes from a wide range of sources, and different contents of pectin exist in fruit and vegetable raw materials such as pomegranate, lemon, apple, grapefruit, hawthorn, cocoa husk, sunflower heads, sugar beet, pumpkin, watermelon, pears and potato pulp [20,21].”
- Podetti, C.; Riveros-Gomez, M.; RománM.C.; Zalazar-García, D.; Fabani, M.P.; Mazza, G.; Rodríguez, R. Polyphenol-enriched pectin from pomegranate peel: Multi-objective optimization of the eco-friendly extraction process. Molecules 2023, 28, 7656.
- Chandel, V.; Biswas, D.; Roy, S.; Vaidya, D.; Verma, A.; Gupta, A. Current advancements in pectin: Extraction, properties and multifunctional applications. Foods2022, 11, 2683.
Thank you again for reviewing and considering our manuscript for publication. We look forward to hearing from you soon.
With best regards
Sincerely yours
Lei Chen, Ph. D
